# Isolation of Microalgae from Mediterranean Seawater and Production of Lipids in the Cultivated Species

**DOI:** 10.3390/foods9111601

**Published:** 2020-11-04

**Authors:** Imane Haoujar, Francesco Cacciola, Manuel Manchado, Jamal Abrini, Mohammed Haoujar, Kamal Chebbaki, Marianna Oteri, Francesca Rigano, Domenica Mangraviti, Luigi Mondello, Adil Essafi, Hicham Chairi, Nadia Skali Senhaji

**Affiliations:** 1Laboratory of Biotechnology and Applied Microbiology, Department of Biology, Faculty of Sciences of Tetouan, Abdelmalek Essaadi University, 93000 Tetouan, Morocco; abrinij@hotmail.com (J.A.); senhajin@hotmail.com (N.S.S.); 2Specialized Center in Zootechnics and Marine Aquaculture Engineering, National Institute of Fisheries Research, 93200 M’Diq, Morocco; chebbakikamal@yahoo.fr; 3Department of Biomedical, Dental, Morphological and Functional Imaging Sciences, University of Messina, 98125 Messina, Italy; 4IFAPA El Toruño, 11500 El Puerto de Santa Maria, Spain; manuel.manchado@juntadeandalucia.es; 5IAV Hassan II, Madinat Al Irfane 6202 Instituts, 10101 Rabat, Morocco; haoujartopographie@gmail.com; 6Department of Veterinary Sciences, University of Messina, Viale Annunziata, 98168 Messina, Italy; moteri@unime.it; 7Department of Chemical, Biological, Pharmaceutical and Environmental Sciences, University of Messina, 98168 Messina, Italy; frigano@unime.it (F.R.); dmangraviti@unime.it (D.M.); lmondello@unime.it (L.M.); 8Chromaleont s.r.l., c/o Department of Chemical, Biological, Pharmaceutical and Environmental Sciences, University of Messina, 98168 Messina, Italy; 9Department of Sciences and Technologies for Human and Environment, University Campus Bio-Medico of Rome, 00128 Rome, Italy; 10BeSep s.r.l., c/o Department of Chemical, Biological, Pharmaceutical and Environmental Sciences, University of Messina, 98168 Messina, Italy; 11National Aquaculture Development Agency, 10100 Rabat, Morocco; adil.essafi@yahoo.com; 12Laboratory of Biological Engineering, Agri-Food and Aquaculture, Department of Biology, Faculty poly-disciplinary of Larache, Abdelmalek Essaadi University, 92000 Larache, Morocco; hicham.chairi@yahoo.fr

**Keywords:** microalgae, isolation, identification, lipid productivity, HPLC-MS

## Abstract

Isolation and identification of novel microalgae strains with high lipid productivity is one of the most important research topics to have emerged recently. However, practical production processes will probably require the use of local strains adapted to commanding climatic conditions. The present manuscript describes the isolation of 96 microalgae strains from seawater located in Bay M’diq, Morocco. Four strains were identified using the 18S rDNA and morphological identification through microscopic examination. The biomass and lipid productivity were compared and showed good results for *Nannochloris* sp. (15.93 mg/L/day). The lipid content in the four species, namely *Nannochloropsis gaditana*, *Nannochloris* sp., *Phaedactylum tricornutum* and *Tetraselmis suecica*, was carried out by high performance liquid chromatography coupled to mass spectrometry (HPLC-MS ) highlighting the identification of up to 77 compounds.

## 1. Introduction

Aquatic microalgae are photosynthetic microorganisms that live with a variety of other species and meet different ecological requirements [1]. They represent one of the first forms of life on earth, and they have been found in oceans for more than 3 billion years since terrestrial environmental components were installed [2,3]. Fifty thousand microalgae species with diverse groups like Cyanobacteria, prokaryotic and eukaryotic microalgae have been discovered in oceans and freshwater lakes, ponds, and rivers around the world, however, only thirty thousand of them have been analyzed [1]. Thanks to their biological property, microalgae can be used as a new source of compounds in several biotechnological applications, including wastewater treatment [4], biodiesel production [5], and as supplements for human and animal dietary [6,7].

A large amount of funding has recently been invested to select the best species of microalgae with high bioactive metabolites [8]. Microalgae represent several sources of bioactive compounds, such as polyphenols, carotenoids, polysaccharides, omega-3, fatty acids, and polyunsaturated fatty acids (PUFA) [9,10,11,12]. The lipid concentrations in microalgae are between 20% and 70%, and the fatty acid composition in algal cells is highly dependent on genetic and phenotypic agents, including environmental and culture conditions [13]. Large scale lipid production will command the use of competitive species that are easy to grow and adapt to local environmental conditions. Isolating strains of microalgae with rapid growth, high intrinsic lipid content, and high biomass productivities is a primary necessity [14,15,16]. The quantity of total lipids in the form of glycolipid, phospholipid, and neutral lipid is varied considerably among and within groups of microalgae [6]. Many prior studies have identified the percent of omega-3 fatty acids between 30 and 40% of their total fatty acids in several species of microalgae like *Nannochloropsis* sp. (EPA), *Schizochytrium limacinum* (DHA), and *Phaeodactylum tricornutum* [17].

As a consequence, microalgae have great potential in the human diet as supplements for the treatment of physiological aberrations, prevention management, and used as synthetic dietary supplements to providing sustainable natural resources [18]. 

The isolation and identification of algal species from a natural environment is a well-established procedure. Each species has different growing conditions, including several regulations and key physic-chemical factors controlling growth and development such as temperature, pH, salinity, and silicate for diatoms [19,20].

The main objective of the present study was to evaluate the effect of the dilution technique and agar plate to separate and isolate different microalgae from Bay M’diq, Morocco. Biomass productivity, lipid productivity, and lipids content of some isolated microalgae were evaluated. Moreover, the 18SrDNA encoding gene of the microalgal isolates was sequenced to confirm the identification of the isolates. The ultra-high performance liquid chromatography coupled to mass spectrometry (UHPLC-MS) has been successfully used for the analysis of several lipid classes such as triacylglycerols (TGs), glycolipids, and phospholipids [21].Ahigh chromatographic resolution was achieved by serially coupling two narrow-bore partially porous columns [22]. A reversed-phase stationary phase, viz.octadecylsilica (C18), was used in order to obtain a good separation of lipids, according to their different hydrophobicity. As for MS detection, a single quadrupole was used as an analyzer and the atmospheric pressure chemical ionization (APCI) interface was used as an ion source. The APCI, different from the most common electrospray (ESI), is able to generate some in-source fragmentation, useful for structure elucidation, especially when a single quadrupole is employed as the MS analyzer and tandem MS experiments cannot be carried out [23,24,25].

## 2. Materials and Methods

### 2.1. Sampling and Isolation

Different species of microalgae were collected from seawater at Bay of M’diq, which is located in the north-western part of Morocco, between Sebta (35°54′ N, 5°1′10′’W) and Capo Negro (35°40′ N, 16′40′’ W). Samples were selected from three collection sites: (i) proximity to a fish farm of the bay; (ii) off the coast of Martil, and (iii) at Kabila Port (Figure 1). A volume of 1 L of seawater sample was taken using clean bottles at a depth of 0.5 m and then stored in cool boxes for transportation to the laboratory.

Seawater samples were taken from December 2017 to February 2018 in a regular manner with constant frequencies with nine samples per month.

Once in the laboratory, the samples are purified using two methods; they were inoculated on an agar plate and in a liquid medium (Guillard F/2) that contained: NaNO_3_ 8.82 × 10^−4^ M; NaH_2_PO_4_·H_2_O 3.62 × 10^−5^ M; Na_2_SiO_3_·9H_2_O 1.06 × 10^−4^ M; FeCl_3_·6H_2_O 1.17 × 10^−5^ M; Na_2_EDTA·2H_2_O 1.17 × 10^−5^ M; MnCl_2_·4H_2_O 9.10 × 10^−7^ M; ZnSO_4_·7H_2_O 7.65 × 10^−8^ M; CoCl_2_·6H_2_O 4.20 × 10^−8^ M; CuSO_4_·5H_2_O 3.93 × 10^−8^ M; Na_2_MoO_4_·2H_2_O 2.60 × 10^−8^ M; Thiamin HCl (Vit. B1) 2.96 × 10^−7^ M; Biotin (Vit. H) 2.05 × 10^−9^ M; and Cyanocobalamin (Vit. B12) 3.69 × 10^−10^ M [26].

For those isolated on an agar plate, the first group of samples was filtered through a series of membranes of decreasing mesh (33, 20, and 0.45 µm). Each membrane was directly plated on agar plates containing F/2medium solidified with 1.5% of agar and incubated in a light chamber at two temperatures (20 and 26 °C). After growth, different colonies were transferred to tubes of 8 mL [27]. However, the second group of samples was isolated by successive dilutions, starting with 1 mL of sample to 10 mL of F/2 medium in order to transfer only one cell into a test tube, thereby establishing a single-cell isolate [28,29]. This procedure of dilution was repeated with serial dilutions from 10^−2^ to 10^−10^ until obtained unicellular tubes medium [30,31] (Figure 2).

During the isolation process, the F/2 medium using in this experiment was divided into two groups; one with and the other one without silicate, to select only strains that required this nutrient. All strains incubated at two temperatures (20 and 26 °C) to evaluate the effect of temperature on cell growth; All cultivations were alimented with atmospheric CO_2_. The light was provided by warm white fluorescent bulbs at 25 W/m^2^ and operated on a light/dark cycle of 12/12 h.

After cell growth, all tubes were inoculated by transferring to 125 mL flasks add by 70 mL of F/2 medium, then incubated with a photoperiod of 18/6 h.

### 2.2. Strain Identification

The morphological identification of different isolated strains was carried out by microscopic observation. At the end, for molecular identification, DNA was extracted using 1030 mg of microalgal dried biomass.

Samples were homogenized using the FastDNA kit for 40 s at speed setting 5 in the Fastprep FG120 instrument (Bio101, Inc., Vista, CA, USA) but using the reagents of the ISOLATE II Genomic DNA Kit (Bioline) kit following the manufacture’s protocols. DNA was quantified spectrophotometrically using the Nanodrop ND8000.

PCR amplifications were carried out using the Platinum Multiplex Master Mix (Thermofisher) in a 25 µL final volume containing 20 ng DNA, 300 nM each of specific forward and reverse primers, and 12.5 µL of Enzyme Premix (Thermofisher). The amplification protocol used was as follows: initial 11 min denaturation and enzyme activation at 95 °C, 30 cycles of 20 s at 95 °C, 1 min at 56 °C (60 °C) and 2 min at 72 °C and final extension step of 10 min at 72 °C. The 18S rDNA primers used were 18SF and euk516R [32,33] and for plastidic gene ribulose-1,5-bisphosphate carboxylase/oxygenase large subunit (rbcl) TetraRBCL_F and R [34] (Table 1). The PCR products were electrophoresed on a 2% agarose gel after staining with ethidium bromide and visualized via ultraviolet transillumination. Following the PCR reaction, free dNTPs and primers were removed using the PCR product purification kit (Marlingen Bioscience, Ijamsville, MD, USA). The cycle sequencing was performed with the Bigdye Terminator v3.1 kit (Applied Biosystems, Foster City, CA, USA). All sequencing reactions were performed according to the manufacturer’s instructions using the ABI3130 Genetic Analyzer (Applied Biosystems). The 18S rDNA and rbcl sequences were used in a BLAST search in order to retrieve the most closely related sequences.

### 2.3. Lipid Analysis

#### Extraction and Measurement of Lipid Contents

The biomass was harvested at a stationary phase growth (after 15 days of cultivation) by centrifugation (at 4400 rpm for 15 min) and lyophilization for 12 h [30].

The evaluation of lipids fraction from selected microalgae was carried out using the Folch method [35]: 200 mg of lyophilized biomass was extracted in triplicate for 30 min with a chloroform/methanol (2:1, *v*/*v*) mixture at 25 °C under agitation. The procedure was repeated three times. The organic phase was centrifuged at 3000 RCF for 15 min. The sample was then filtered using Whatman N°1 filter paper in a funnel, collected in a flask, and evaporated at 40 °C.

The productivity of biomass was calculated from the following equation: PB (mg/L/day) = CB/T; where CB (mg/L) was the concentration of biomass from the beginning until the end of the cultivation, and T was the duration of cultivation (15 days).

In addition, the lipid productivity of each sample was determined by Li et al.’s (2008) equation: PL = LT/(V culture × T) and % lipids = (LT/(CB × V culture)) × 100; where PL was the lipid productivity (mg/L/day), LT was the total lipids (mg), T was the duration of the experiment, and V culture was the volume [36,37].

### 2.4. HPLC Analysis

HPLC-MS analyses of the lipid contents from the obtained residue dissolved in chloroform were performed on a Shimadzu Nexera LC-30A system (Shimadzu, Kyoto, Italy), consisting of a CBM-20A controller, two LC-30AD dual-plunger parallel flow pumps, a DGU-20A5 degasser, a CTO-20A oven, and a SIL-30AC autosampler. The HPLC system was coupled with an LCMS-2020 single quadrupole mass spectrometer equipped with an APCI interface (Shimadzu, Kyoto, Japan). Chromatographic separation was achieved on two serially coupled Ascentis Express C18 columns 100 × 2.1 mm L × I.D., 2.7 µm d.p. (Merck Life Science, Merck KGaA, Darmstadt, Germany), and the injection volume was 10 µL. A linear gradient of ACN/water (80:20, *v*/*v*) (A) and IPA (B) was run at a mobile phase flow rate of 500 µL/min: 0–105 min, 0–50% B (hold for 5 min). MS parameters were as follows: m/z range, 150–1200; ion accumulation time, 0.6 s; nebulizing gas (N_2_) flow rate, during gas flow rate, 2 L/min; detector voltage, 4.5 kV; interface temperature, 450 °C; CDL temperature, 250 °C; block temperature, 300 °C.

### 2.5. Statistical Analysis

All statistical tests were performed using SPSS statistical software (SPSS software version 16.0 IBM). The different medians of results were analyzed by one-way ANOVA with a significance level of *p* < 0.05 and compared by Tukey’s TSD method [38].

## 3. Results and Discussion

### 3.1. Isolation of Native Microalgae

The sampling and isolation procedures used in this experiment were successful in establishing a culture collection of ninety-six local microalgae. The microscopic analysis showed that the majority were green algae (Chlorophyta), followed by diatoms and finally cyanobacteria with the lowest number. Sixty-four microalgae were isolated using successive dilution and thirty-two microalgae by inoculation on agar plate (Figure 3A). Following the analysis of variance, ANOVA one-way showed that the difference was insignificant (*p* > 0.05) between the two isolation processes. The two processes used in this experiment turned out to be reproducible to purify microalgal cells. The use of agar plate for isolation was the preferred method for Coccoid and the most soil algae since it represents an axenic culture for direct established without further treatment [28,31].

Seventy and twenty-six species were isolated using F/2 culture medium with and without silicate, respectively. The highest number of species were isolated in the culture medium with silicate. The highest number of isolated species was obtained in December 2017 using F/2 medium added by silicate (Figure 3B). In addition, the statistical analysis using One-way ANOVA (Table 2) revealed that the difference between the two groups was highly significant (*p* < 0.01). Therefore, the silicate nutrient added to the F/2 medium had a great impact in regulating the cell growth [39].

Following previous results, the highest number of species, isolated using a culture medium added by silicate, may be explained by the dominance of diatoms. Moreover, Egge and Aksnes (1992) confirmed that diatom strains represent 70% of the cell numbers by using a concentration of 2 mM silicate [40].

Temperature is one of the key criteria for the growth of microalgae and directly acts on the linear and exponential growth of microalgae species [41,42]. In this experiment, fifty species and forty-six strains were isolated at 26 °C and 22 °C, respectively (Figure 3C). Our results were confirmed by Ahlgren’s study [41], which showed that the range of temperatures between 16–27 °C was determined as optimal for algal growth rates. Consequently, the species of microalgae were flexible and adaptable with the temperatures tested and able to produce several physiological and biochemical reactions [41,42,43,44]. The insignificant difference (*p* = 0.823) between the temperatures confirmed the possibility of growth cells at 26 °C and 22 °C.

### 3.2. Molecular Identification of Native Microalgae

Four species of microalgae collected from different seawater habitats were identified by microscopic morphological examination based on the form of their cells (Figure 4). Subsequently, two molecular markers were considered to ensure their taxonomic group (Figure 5). The isolated microalgae 1, 4, 5, and 7 were closely related to *Nannochloropsis gaditana* Lubián (MN625926), *Phaeodactylum tricornutum* Bohlin (MN625939), *Nannochloris* sp. KMMCC161 Naumann (MN625923) and *Tetraselmis suecica* Butcher (MN625941) deposited in the NCBI database under GenBank with mentioned accession numbers in parentheses.

### 3.3. Biomass and Lipid Productivity

One of the key criteria to select microalgae for lipid applications is high intracellular lipid content [45]. Growth and biomass productivity are the most studied parameters to determine the suitable microalgae able to cultivate and using for commercial algal production. In our experiment, a significant difference in lipid levels was determined. The highest biomass productivity was reported in *N. gaditana* then, *Nannochloris* sp., *P. tricornutum,* and *T*. *suecica* with 97.09, 96.92, 24.47, and 12.54 mg/L/day, respectively. The highest and the lowest levels of lipid productivity were respectively obtained for *Nannochloris* sp., and *T*. *suecica* species that were measured to be 15.93 and 0.92 mg/L/day (Table 3). Growth rate and biomass productivity directly affect the lipid content [46]; lower biomass productivity can negatively affect productivity even if the lipid content is high [47].

The total lipid contents of the microalgae cultured under our experimental growth conditions ranged from 18.28% to 47.43% of their dry weight. *P. tricornutum* and *Nannochloris* sp., showed the highest lipid content at 47.43% and 41.08%, while *T. suecica* and *N*. *gaditana* have the lowest lipid content at 18.28 and 14.28%, respectively. *Dunaliella tertiolecta* species isolated from Morocco by El Arroussi et al. (2017) had a lipid content of 21%, which confirmed that the lipid productivity was greatly affected by cultivation conditions; salinity, light intensity, temperature, pH of medium, and composition of culture medium used [21,45]. In addition, many studies have confirmed that the limited concentration of nitrogen favored the accumulation of lipids in *Chlorella* species [46,47,48,49].

### 3.4. HPLC Analysis of Lipid Contents

LC-MS analysis was applied to separate and identify the lipid fraction in the four isolated microalgae. The separation of lipids classes reported in Figure 6 confirmed a good separation of neutral and polar lipids. Table 4 lists the details of different lipid classes tentatively identified in all algal samples confirmed by literature data [23,25,50,51]. The identification was carried out with the support of LIPID MAPS Lipidomics Gateway (https://www.lipidmaps.org/), which allowed the identification of a total of 77 compounds in all the selected microalgae species. The lipid composition was constituted basically by mono-, di- and triglycerides (MG, DG, and TG), sulfoquinovosyldiacylglicerols (SQDG), mono- and digalactosyldiacylglycerols (MGDG and DGDG), phospholipids (PLs) and carotenoids (xanthophylls and chlorophylls) in the four microalgal isolates. The lipid fraction of microalgae can also include glycolipids, sterols, carotenoids, chlorophylls, and phospholipids. Algae can synthesize fatty acids to product their membrane lipids in the optimal growth conditions, which constitute phospholipids and glycolipids. However, in conditions of stress, algae alter their biosynthetic pathways and generate neutral lipids to store energy in the form of TGs [52,53,54]. Notably TGs are significantly produced by the environmental microalgal isolates as demonstrated by the relatively intense peaks of the chromatograms. In fact, intense peaks were identified in *N*. *gaditana* and *Nannochloris* sp. with TGs C17:0LAr/C17:0OEp—OOP/SLnP and in *T*. *suecica* with TGs (OOP/PPoG/PoOL), while in *P. tricornutum*, glycolipids SQDG (34:4) occurred as the highest peaks. However, previous studies have demonstrated that some *Tetraselmis* sp., *Nannochloris* sp., *Phaeodactylum tricornutum*, and *Nannochloropsis gaditana* species can produce more lipids under certain stressed conditions [32,47,49,50].

The comprehensive lipid profiles obtained for the four strains showed their higher potential of glycolipids, phospholipids, and triacylglycerols to be an ideal source of food additives, ingredient for functional foods, and nutraceuticals applications [31,51].

## 4. Conclusions

This study reports the identification of four strains of microalgae from Moroccan Mediterranean seawater, which might be suitable for lipid production to be potentially used as supplement aquatic and animal feed rich in PUFA and other food products with higher omega-3 fatty acids content. In particular, the data from this study showed that *Nannochloris* sp., had the highest lipid productivity of 15.93 mg/L/day. Furthermore, the HPLC-MS analysis showed their highest content of lipidic molecules (77 in total) and might be useful as dietary supplements or biofuels feedstock.

## Figures and Tables

**Figure 1 foods-09-01601-f001:**
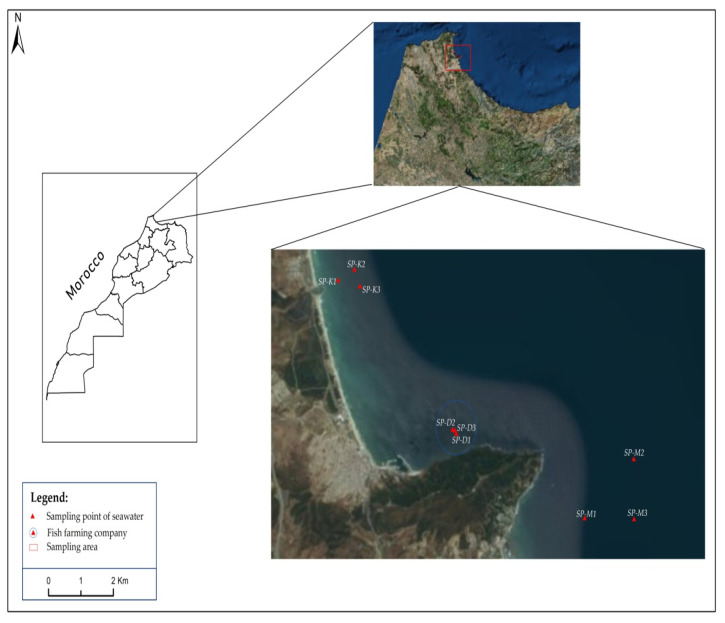
Distribution map of sampling points in M’diq Bay, Morocco.

**Figure 2 foods-09-01601-f002:**
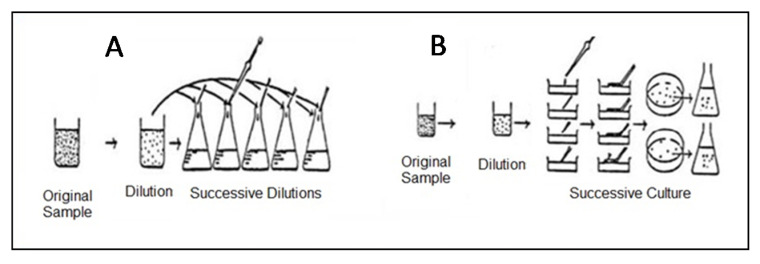
Method of cell isolation with (**A**) successive dilution, and (**B**) inoculation on an agar plate.

**Figure 3 foods-09-01601-f003:**
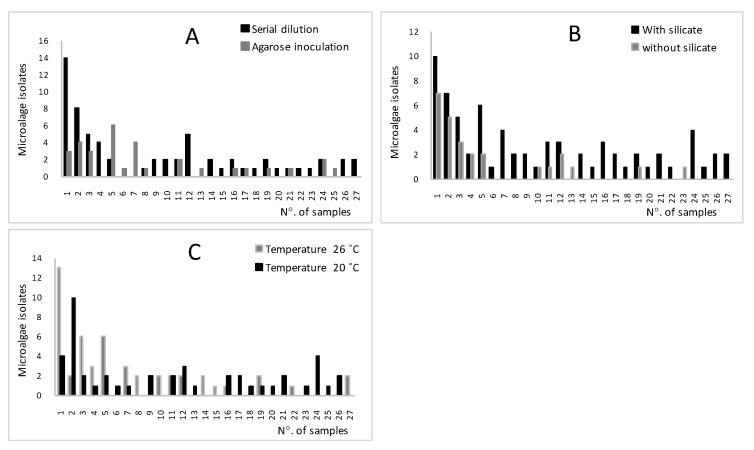
Strains curves for isolated microalgae using successive dilution, (**A**) inoculating on an agar plate, (**B**) in F/2 with or without silicate and (**C**) incubation at 20 °C and 26 °C.

**Figure 4 foods-09-01601-f004:**
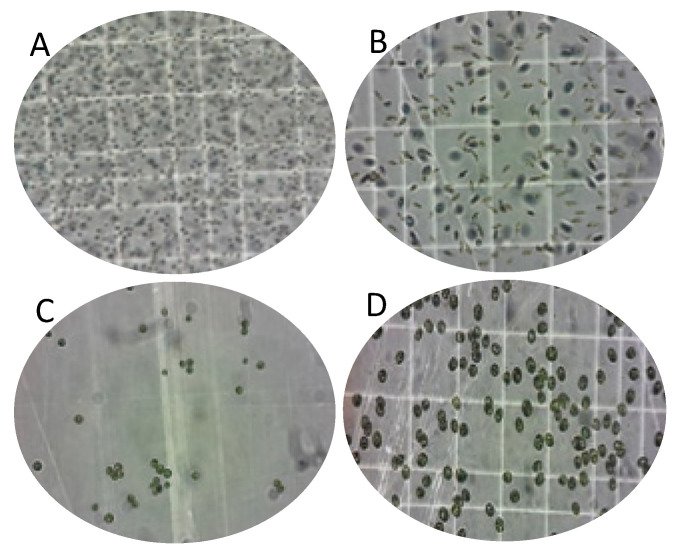
Light microscopic observation of isolated microalgae: (**A**) *Nannochloropsis gaditana,* (**B**) *Phaeodactylum tricornutum,* (**C**) *Nannochloris* sp., and (**D**) *Tetraselmis suecica*.

**Figure 5 foods-09-01601-f005:**
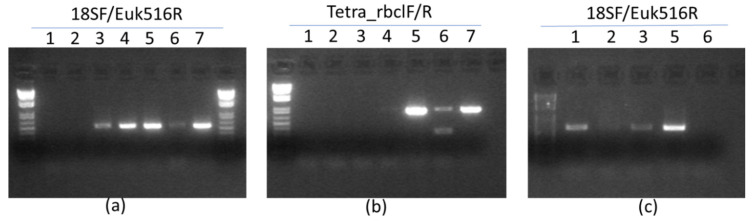
Molecular identification of selected microalgae. 1 to 7: number of strains. (**a**,**b**): annealing T° of 56 °C. (**c**): annealing T° of 60 °C.

**Figure 6 foods-09-01601-f006:**
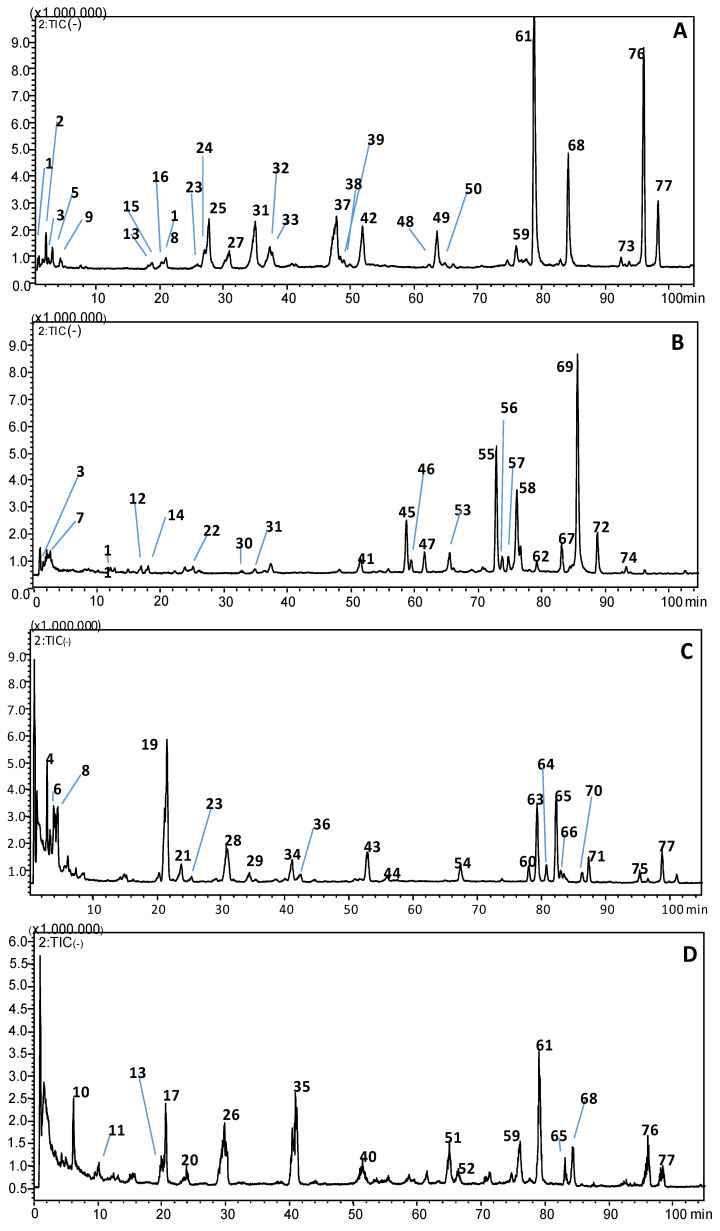
Base peak chromatograms of lipids extract of four isolated microalgae. (**A**) *N*. *gaditana*; (**B**); *T*. *suecica*; (**C**) *P*. *tricornutum*; (**D**) *Nannochloris* sp. For experimental conditions, see the text. For peak identification, see Table 4.

**Table 1 foods-09-01601-t001:** List of primers used in this study, including the primer sequences, amplicon length, annealing temperature, and the sequencing success rate for four strains tested.

Primer	Molecular Marker	Sequence	Annealing T°	Reference
18SF	18S rDNA	AACCTGGTTGATYCTGCCAG	56 °C, 60 °C	[32]
Euk516r	18S rDNA	ACCAGACTTGCCCTCC	56 °C, 60 °C	[33]
Tetra_rbcL_F	rbcl	GKACTTGGACAACTGTATGGACKGATGGT	56 °C	IFAPA
Tetra_rbcL_R	rbcL	GRTCTTTTTCWACRTAAGCATCACGCATTA	56 °C	IFAPA

**Table 2 foods-09-01601-t002:** ANOVA-test statistics values of each isolated microalgae.

One-Factor ANOVA					
Parameter	Source	df	Sum of Squares	F	P
Silicate	Inter-group	1	35.85	9.07	0.004 *
	Intra-group	52	205.48		
	Total	53	241.33		

* The average difference is significant at the 0.05 level.

**Table 3 foods-09-01601-t003:** Biomass and Lipid Content of Selected Microalgae Species in Batch Culture (400 mL) after 15 Days of Growth in F/2 Medium.

Species	Weight of dry biomass (mg/400 mL)	Concentration of biomass (mg/L)	Productivity of biomass (mg/L/day)	Total lipids (mg/400mL)	Lipids (%)	Lipid productivity (mg/L/day)
*P*. *tricornutum*	146.82 ± 1.1 ^b^	367.05 ± 0.15 ^b^	24.47 ± 0.15 ^b^	27.85 ± 0.1 ^b^	47.43 ± 0.68 ^d^	4.64 ± 0.1 ^b^
*T*. *suecica*	75.25 ± 0.2 ^a^	188.13 ± 0.9 ^a^	12.54 ± 0.03 ^a^	5.50 ± 0.04 ^a^	18.28 ± 0.7 ^a^	0.92 ± 0.02 ^a^
*N*. *gaditana*	582.53 ± 0.15 ^c^	1456.33 ± 1.43 ^c^	97.09 ± 0.68 ^c^	33.29 ± 0.8 ^c^	14.28 ± 0.32 ^b^	5.55 ± 0.01 ^b^
*Nannochloris* sp.,	581.52 ± 1.4 ^c^	1453.8 ± 1.3 ^c^	96.92 ± 0.72 ^c^	95.58 ± 0.81 ^d^	41.08 ± 0.26 ^c^	15.93 ± 0.9 ^c^

Values are medians of three repetitions ± standard deviation. The different letters (a,b,c,d) indicate significant differences between species (Tukey HSD, *p* < 0.05).

**Table 4 foods-09-01601-t004:** Compounds tentatively identified in the lipid fraction.

Peak	Compounds	[M+H]^+^	[M+H]^−^	*N*. *gaditana*	*T*. *suecica*	*P*. *tricornutum*	*Nannochloris* sp.,
1	MG 20:0	369.5		+	-	-	-
2	MG 16:0	313.2		+	-	-	-
3	DG (36:4)	599.5		+	+	-	-
4	PG (34:3)		762.3	-	-	+	-
5	DG (34:2)	617.3		+	-	-	-
6	SQDG (32:1)	549.5	791.5	-	-	+	-
7	DG (36:2)	603.4		-	+	-	-
8	DG (32:1)	549.5		-	-	+	-
9	Neoxanthin	601.7		+	-	-	-
10	DG (32:0)	551.4		-	-	-	+
11	PG (34:4)		741.4	-	+	-	+
12	PG (34:3)		762.3	-	+	-	-
13	MGDG (34:5)	766.5		+	-	-	-
14	DG (32:5)	583.6		-	+	-	-
15	TG (ArArO/SArEp)	928.5	909.7	+	-	-	-
16	PE (34:3)		762.4	+	-	-	-
17	DG (32:4)	585.5		-	-	-	+
18	DGDG (36:6)	954.6	935.4	+	-	-	-
19	SQDG (34:4)		813.6	-	-	+	-
20	DGDG (34:4)	930.6		-	-	-	+
21	DGDG (36:6)	954.6	935.4	-	-	+	-
22	Antheraxanthin	585.9		-	+	-	-
23	PC (36:1)	788.5	812.7	+	-	+	-
24	SQDG (34:0)		821.5	+	-	-	-
25	PI (36:4)		857.5	+	-	-	-
26	trans-Lutein	569.9		-	-	-	+
27	TG (C20:3LL)	907.6		+	-	-	-
28	SQDG (34:3)		815.5	-	-	+	-
29	TG (20:3LL)	907.5		-	-	+	-
30	PG (34:1)		766.3	-	+	-	-
31	PC (36:3)		784.5	+	+	-	-
32	MGDG (36:5)	794.5		+	-	-	-
33	MGDG (34:6)	764.5		+	-	-	-
34	MGDG (36:6)	792.5		-	-	+	-
35	PG (36:5)		786.5	-	-	-	+
36	PG (34:1)		766.3	-	-	+	-
37	TG (SOAr)	908.6		+	-	-	-
38	PE (39:6)		776.5	+	-	-	-
39	PE (38:3)		768.5	+	-	-	-
40	PG (36:4)		788.4	-	-	-	+
41	DGDG (34:2)	934.5	915.6	-	+	-	+
42	PC (38:3)	814.5		+	-	-	-
43	PG (34:0)		768.5	-	-	+	-
44	SQDG (34:1)		819.4	-	-	+	-
45	PC (38:5)	808.8		-	+	-	-
46	DGDG (36:1)	964.7		-	+	-	-
47	hydroxychlorophyllide b	645.1		-	+	-	-
48	PC (33:2)		744.5	+	-	-	-
49	PE (38:5)		764.5	+	-	-	-
50	b-carotene	537.9		+	-	-	-
51	TG (LnLnPo)	849.4		-	-	-	+
52	MGDG (34:2)	772.5	753.6	-	-	-	+
53	TG (LnGG/OGLn/SGAr)	936.8		-	+	-	-
54	PC (33:1)–PC (O-16:0/18:1)	746.5		-	-	+	-
55	TG (GEpD)	981.5		-	+	-	-
56	PI (40:8)		905.5	-	+	-	-
57	TG (GLL)	909.5		-	+	-	-
58	TG (ArArAr)	950.5		-	+	-	-
59	PC (32:2)		730.5	+	-	-	-
60	TG (OLC18:4)	877.5		-	-	+	-
61	TG (C17:0LAr/C17:0OEp–OOP/SLnP)	893.6–859.8		+	-	-	-
62	TG (EpC18:4C18:4)	893.4		-	+	-	-
63	TG (GLL/LLnA/OLnG)	909.5		-	-	+	-
64	TG (EpEpL)	923.5		-	-	+	-
65	PI (38:1)		892.6	-	-	+	-
66	TG (SSS)	891.4		-	-	+	-
67	Anhydroeschscholtzxanthin		529.3	-	+	-	+
68	TG (C18:4C18:4Ep)	893.6		+	-	-	+
69	TG (OOP/PPoG/PoOL)	925.6		-	+	-	-
70	PC (44:5)		890.3	-	-	+	-
71	TG (EpC18:4C18:4)	893.6		-	-	+	-
72	TG (ArArLn/OEpEp/ArEpL)	925.6		-	+	-	-
73	TG (SOO/SSL/PLA)	887.5		+	-	-	-
74	TG (MMDh)	823.5		-	+	-	-
75	TG (SOO/SSL/PLA)	887.5		-	-	+	-
76	TG (OOMo/LnLn18:4)	871.4		+	-	-	-
77	Pheophytin a (C_55_H_74_N_405_)	871.4	870.6	+	-	+	+

Note: MG, monoglyceride; DG, diglyceride; TG, triglyceride; SQDG, sulfoquinovosyldiacylglicerol; MGDG, monogalactosyldiacylglycerol; DGDG, digalactosyldiacylglycerol; PG, phosphatidylglycerol; PE, phosphatidylethanolamine; PC, phosphatidylcholine; PI, phosphatidylinositol. M = C14:0; Mo = C14:1; P = C16:0; Po = C16:1; S = C18:0; O = C 18:1; L = C18:2; Ln = C18:3; A = C20:0;0020G = C20:1; Ar = C20:4; Ep = C20:5; B = C22:0; D = C22:5; Dh = C22:6.

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
