# Peer review of "Isolation of Microalgae from Mediterranean Seawater and Production of Lipids in the Cultivated Species"

_foods, 2020, doi:10.3390/foods9111601_

Round 1

Reviewer 1 Report

Foods-975415

Isolation of microalgae from Mediterranean seawater and production of lipids in the cultivated species

The authors described the isolation of 96 samples of marine microalgae strains from places close to Morocco and optimized the isolation conditions. In four cases they also identified the isolated species. For these four species they also measured growth rate and lipid content and finally detected the presence of 77 lipid classes by HPLC-MS. Especially this extensive work of lipid classification makes the project valuable.

The work makes a very professional impression presenting a large number of solid data.

  1. Line 38: Please, write “Four strains were identified”. The present “Selected strains” confused me reading the manuscript.
  2. Line 42: The word species suggests that 77 microalgae species were identified. “Compounds” or “lipid classes” as in Table 4 might be better. See also line 236.
  3. Line 53: Correct language
  4. Line 56: this is more spoken English. Please, use something like “a large number of”
  5. Line 72: Correct language
  6. Line 75: this is spoken English, better perhaps “good/ effective” separation
  7. Line 76: “the” is not required here
  8. Line 96: what are the first and the second group?
  9. Line 100: Please, give the composition of the F/2 medium here (in molar concentrations) and, if possible, reference.
  10. Line 124: After staining with ethidium bromide?
  11. Table 1: If the primers were not designed here (what I assume), then reference should be given.
  12. Lines 141 and 145: This equation makes only sense if there is a linear increase. Is this growth not exponential?
  13. Line 171: Remove “According to the results”.
  14. Line 175: Is there no overlapping at all between species isolated in culture with and without silicate?
  15. Line 177: December of which year? Or is this the experience from several years: Please, make this clear.
  16. Line 224: Correct language
  17. Line 237-247: This is a little bit too short to mention what in the project was discovered. I would like to learn a little bit more about the importance of the different lipid classes and how far they are distributed in other organisms.
  18. Conclusions, line 256-261: These are hardly any conclusions. The authors should make some conclusions from the results and not repeat the results here.

Author Response

Authors comments to respond to the Reviewer 1

Comment 1: Line 38: Please, write “Four strains were identified”. The present “Selected strains” confused me reading the manuscript.

Authors’ Response: We thank the reviewer for pointing this out. We have modified “selected strains” by “four strains were identified” (Line 38).

Comment 2: Line 42: The word species suggests that 77 microalgae species were identified. “Compounds” or “lipid classes” as in Table 4 might be better. See also line 236.

Authors’ Response: Thank you very much for your observation. We have modified “species” to “compounds” (Line 42).

Comment 3: Line 53: Correct language

Authors’ Response: Thank you very much for your observation. We have corrected the English language in Line 53-54.

Comment 4: Line 56: this is more spoken English. Please, use something like “a large number of”

Authors’ Response: We thank the Reviewer for pointing this out. We have now followed the suggestion and we have placed the “a lot of funding has been invested” to “a large amount of funding has been invested” (Line 56).

Comment 5: Line 72: Correct language

Authors’ Response: We thank the Reviewer for this important examination. We have corrected the language in the (Line 77-82).

Comment 6: Line 75: this is spoken English, better perhaps “good/ effective” separation

Authors’ Response: We thank the Reviewer for this suggestion. We have inserted “good separation” to the Line 84

Comment 7: Line 76: “the” is not required here

Authors’ Response: Thank you very much for your observation. We have removed “the” from the Line 85.

Comment 8: Line 96: what are the first and the second group?

Authors’ Response: We thank the Reviewer for pointing this out. The first group is corresponding to the samples that were inoculated on agar plate, while the second group is for samples that were treated by successive dilutions in liquid medium (Line108).

Comment 9: Line 100: Please, give the composition of the F/2 medium here (in molar concentrations) and, if possible, reference.

Authors’ Response: We thank the Reviewer for this important suggestion. We have added the composition of F/2 medium (in molar concentration) with reference in the Line 103-107.   

Comment 10: Line 124: After staining with ethidium bromide?

Authors’ Response: We thank the Reviewer for this important examination. Yes it was after staining with ethidium bromide (Line 141).

Comment 11: Table 1: If the primers were not designed here (what I assume), then reference should be given.

Authors’ Response: We thank the Reviewer for this important criticism. We have indicated the references of the primers in Table 1:

- for 18SF and Euk516r the references are 32 (Sakata et al., 2005) and 33 (Dı́ez et al., 2001). (Line 139)

- for Tetra_rbcL_F and Tetra_rbcL_R, these primers have already been developed by IFAPA in Cadiz co-author of this work (Prof. Manuel Manchado). These primers have been published in the Report of the Scientific Committee of the Spanish Agency for Food Safety and Nutrition. We have inserted this reference (Below) in the Line 140.

Rodríguez-Ferri, E., Badiola-Díez, J.J., Cepeda-Sáez, A., Domínguez-Rodríguez, L., Otero-Carballeira, A. y Zurera-Cosano, G. Grupo de trabajo. Informe del Comité Científico de la Agencia Española de Seguridad Alimentaria y Nutrición (AESAN) sobre la evisceración de los lagomorfos. Revista del Comité Científico de la AESAN, 2009, 9, 31-38.

Comment 12: Lines 141 and 145: This equation makes only sense if there is a linear increase. Is this growth not exponential?

Authors’ Response: Thank you very much for your observation. Yes in this study we have used the equation to define the linear increase (Line 159, 163)

Comment 13: Line 171: Remove “According to the results”.

Authors’ Response: Thank you. We have removed “According to the results” from the text.

Comment 14: Line 175: Is there no overlapping at all between species isolated in culture with and without silicate?

Authors’ Response: We thank the Reviewer for pointing this out. No there is no overlapping; we have obtained seventy species using F/2 medium with silicate and twenty-six species by using F/2 medium without silicate (Line 194,195).

Comment 15: Line 177: December of which year? Or is this the experience from several years: Please, make this clear.

Authors’ Response: We thank the Reviewer for this note. We have now followed the suggestion and we have indicated that the samples were taken in December 2017 in the Line 196.

Comment 16: Line 224: Correct language

Authors’ Response: We thank the Reviewer for this suggestion. We have corrected the language in the (Line 241-244).

Comment 17: Line 237-247: This is a little bit too short to mention what in the project was discovered. I would like to learn a little bit more about the importance of the different lipid classes and how far they are distributed in other organisms.

Authors’ Response: We thank the Reviewer for this important criticism. We have now followed his/her suggestion and we have added more information about the importance of the different lipid classes and how far they are distributed in other organisms in the Line 264-268, 275-277.

Comment 18: Conclusions, line 256-261: These are hardly any conclusions. The authors should make some conclusions from the results and not repeat the results here.

Authors’ Response: We thank the Reviewer for this note. We have now followed the suggestion from the reviewer and make some conclusions from the results (Line 286-291).

Reviewer 2 Report

The main aim of the paper is to isolate and identificate novel microalgae strains with high lipid productivity

The results are interesting. However, the introduction is very scarce, and more information is needed. Material and methods need to be improved and a detailed statistical analysis of the data is needed. Discussion should consider previous data and give more information.

Specific comments

Line 47. Introduction should give more information about previous studies on microalgae with examples of level of production of lipids and isolation procedure, not just focus on the present experiment.

Line 85. How samples were taken. Amount of water, deepness, methodology…

Line 93. Only agar? Not other components in the media?

Line 101. Quantity of silicate?

Line 168. List and number of algae isiolated? Proportion of cyanobacteria, green algae…. More information needed.

Line 198. Only  four species? Before there were more than 70 (line 175). Explain difference

Line 201. Latin name with the author. Nannochloropsis gaditana Lubián

Line 228. Statistical analysis needed. Error of the average? Values are statistical different?

Line 252. I would use + in stead of x to note that the compound is present.

Line 247. No comparison with other studies? Are levels normal? There are new compounds? More discussion needed.

Author Response

Authors comments to respond to the Reviewer 2

Comment 1: Line 47. Introduction should give more information about previous studies on microalgae with examples of level of production of lipids and isolation procedure, not just focus on the present experiment.

Authors’ Response: We thank the Reviewer for this important suggestion. We have now followed his/her suggestion and added a paragraph that reported more information about previous studies on microalgae with examples of levels of production of lipids and the importance to isolate the local species of microalgae (Line 56-68).

Comment 2: Line 85. How samples were taken. Amount of water, deepness, methodology…

Authors’ Response: We thank the Reviewer for pointing this out. We have mentioned that strains have been isolated by taken a volume of 1 L of seawater sample from three collection sites : (i) proximity to a fish farm of the bay; (ii) of the coast of Martil, and (iii) of Kabila Port using clean bottles at a depth of 0.5 m and then stored in cool boxes for transportation to the laboratory (Line 95-97)

Comment 3: Line 93. Only agar? Not other components in the media?

Authors’ Response: We thank the Reviewer for pointing this out. We have used the other components of culture medium Guillard F/2 (Line 103-107) to the agar (1.5% ) in order to prepare a solid medium (agar plates) (Line 110).

Comment 4: Line 101. Quantity of silicate?

Authors’ Response: We thank the Reviewer for this important examination. We have added the concentration of silicate (Na2SiO3.9H2O 1.06×10-4 M) that was used in this study in the Line 103.

Comment 5: Line 168. List and number of algae isiolated? Proportion of cyanobacteria, green algae…. More information needed.

Authors’ Response: We thank the Reviewer for this important suggestion. We have now followed his/her suggestion and mentioned the most dominant class of algae in our study (Line 186-188).

Comment 6: Line 198. Only  four species? Before there were more than 70 (line 175). Explain difference

Authors’ Response: We thank the Reviewer for pointing this out. A total of 96 algal cultures were isolated from seawater located in the Bay M’diq, Morocco. Four microalgal isolates were selected based on their morphology, rapid and ease of cultivation under our test conditions. The same reason has been taken into consideration in several studies to select, identify, and evaluate certain species among several isolated [1–3] 

  1. Abdelaziz, A.E.M.; Leite, G.B.; Belhaj, M.A.; Hallenbeck, P.C. Screening microalgae native to Quebec for wastewater treatment and biodiesel production. Bioresour. Technol. 2014, 157, 140–148.
  2. Duong, V.T.; Li, Y.; Nowak, E.; Schenk, P.M. Microalgae isolation and selection for prospective biodiesel production. Energies. 2012, 5, 1835–1849.
  3. Abou-Shanab, R.A.; Matter, I.A.; Kim, S.-N.; Oh, Y.-K.; Choi, J.; Jeon, B.-H. Characterization and identification of lipid-producing microalgae species isolated from a freshwater lake. Biomass Bioenergy. 2011, 35, 3079–3085.

Comment 7: Line 201. Latin name with the author. Nannochloropsis gaditana Lubián

Authors’ Response: We thank the Reviewer for this important suggestion. We have now followed his/her suggestion and we have added the Latin names with authors to the species in the Line 221-223.

Comment 8: Line 228. Statistical analysis needed. Error of the average? Values are statistical different?

Authors’ Response: We thank the Reviewer for this important suggestion. In Table 3, we have presented the data with ± SD (Standard deviation). We have used Tukey’s TSD method to show the significant difference in lipids levels (Line 234,235)

Comment 9: Line 252. I would use + instead of x to note that the compound is present.

Authors’ Response: Thank you very much for your observation. We have modified (×) to (+) in the table 4 to note that the compounds is present (Line 284-285).

Comment 10: Line 247. No comparison with other studies? Are levels normal? There are new compounds? More discussion needed.

Authors’ Response: We thank the Reviewer for this important suggestion. We have now followed his/her suggestion and compared our species with others in other studies (Line 272-277). The lipid compounds obtained by our analysis are the same as those obtained by other studies, which confirms that the lipid content of microalgae is rich and consequently can be used as food supplements.

Round 2

Reviewer 2 Report

The authors made improvements.

Just in table 3 put the letters of statistical significance after the standard error and at the same size of the numbers to make them easier to see.

Author Response

Comment: Just in table 3 put the letters of statistical significance after the standard error and at the same size as the numbers to make them easier to see.

Authors’ Response: We thank the Reviewer for this important suggestion. We have now followed his/her suggestion and we have mentioned the letters of statistical significance after the standard error and at the same size as the numbers in table 3.